



# A new approach to estimate supersaturation fluctuations in stratocumulus cloud using ground-based remote sensing measurements

Fan Yang[1], Robert McGraw[1], Edward P. Luke[1], Damao Zhang[1], Pavlos Kollias[1,2], and Andrew M. Vogelmann[1]

[1]Brookhaven National Laboratory, Upton, NY, USA
[2]School of Marine and Atmospheric Sciences, Stony Brook University, Stony Brook, NY, USA

**Correspondence:** Fan Yang (fanyang@bnl.gov)

**Abstract.** Supersaturation, crucial for cloud droplet activation and condensational growth, varies in clouds at different spatial and temporal scales. In-cloud supersaturation is poorly known and rarely measured directly. On the scale of a few tens of meters, supersaturation in clouds has been estimated from in-situ measurements assuming quasi-steady state supersaturation. Here, we provide a new method to estimate supersaturation using ground-based remote sensing measurements, and results are

compared with those estimated from aircraft in-situ measurements in a marine stratocumulus cloud during the ACE-ENA field campaign. Our method agrees reasonably well with in-situ estimations and it has three advantages: (1) it does not rely on the quasi-steady state assumption, which is questionable in clean or turbulent clouds; (2) it can provide a supersaturation profile, rather than just point values from in-situ measurements; and (3) it enables building statistics of supersaturation in stratocumulus clouds for various meteorological conditions from multi-year ground-based measurements. The uncertainties, limitations and

possible applications of our method are discussed.

## 1 Introduction

Cloud forms under supersaturated conditions when the air contains more water vapor than it can retain. Supersaturation ($s$), varying from less than $0.1\%$ in calm stratiform clouds to larger than $1\%$ in energetic convective clouds, plays a crucial role in cloud droplet formation and growth (Lamb and Verlinde, 2011). $s$ in atmospheric clouds fluctuates on a wide range of

temporal and spatial scales because of turbulence. Previous studies show that the average value of $s$ as well as its fluctuation are important for cloud microphysical processes. For example, the variation of supersaturation in a vertically oscillating cloud parcel can cause droplet deactivation and reactivation (see Yang et al. (2018a) and references therein). Additionally, results from recent laboratory experiments and direct numerical simulations show that stochastic diffusional growth due to supersaturation fluctuations in a turbulent environment can broaden the cloud droplet size distribution, which might be important to drizzle

formation in warm clouds (e.g., Sardina et al., 2015; Chandrakar et al., 2016; Li et al., 2019).





Despite its importance, $s$ is barely known and poorly measured in atmospheric clouds. If we can simultaneously measure water vapor pressure ($e$) and temperature ($T$), we can directly calculate $s$ based on its definition,

$$s = \frac{e}{e_{sat}} - 1, \tag{1}$$

where $e_{sat}$ is the saturated water vapor pressure which depends only on $T$. However, it is extremely difficult to measure $e$ and $T$ precisely in clouds due to the influence of liquid droplets on the measurements (Siebert and Shaw, 2017). For example,
fast-response measurements of water vapor mixing ratio usually rely on the absorption of infrared light, which can be affected by the presence of liquid droplets or the accumulation of a liquid film on the instrument. Temperature measurements can be biased by latent heat of condensation, evaporation, and also ram heating caused by the deceleration of air on the aircraft body (Wendisch and Brenguier, 2013).

Because of the difficulties in using direct measurements, $s$ is usually estimated indirectly based on either (1) cloud conden-
sation nuclei (CCN) - cloud droplet number concentration (CDNC) closure, or (2) the quasi-steady state assumption. For the CDNC closure method, $s$ in the CCN counter is taken to be the effective $s$ in clouds if the number concentrations of CCN and cloud droplets are similar (e.g., Yum et al., 1998). However, this method might overestimate $s$ due to kinetic limitations (Nenes et al., 2001; Yang et al., 2012). Additionally, the effective $s$ for cloud droplet activation can only be positive whereas the $s$ of interest here can be either positive or negative due to turbulence. The second method is to estimate $s$ in clouds based on the
quasi-steady state assumption through the following formula (e.g., Politovich and Cooper, 1988),

$$s \approx s_{qs} = A \frac{w}{N_d \bar{r}}, \tag{2}$$

where $s_{qs}$ is the quasi-steady state supersaturation, $A$ is a parameter that depends on temperature and pressure (see Equation A1 in Appendix), $w$ is vertical air velocity, $N_d$ is the cloud droplet number concentration, and $\bar{r}$ is the mean cloud droplet radius. This method relies on the quasi-steady state assumption in which the source/sink of water vapor in an adiabatic air parcel due to vertical motion is roughly balanced with the sink/source of water vapor due to condensation/evaporation. However, this
assumption might not be valid in clean (low $N_d$) and/or vigorous (large $w$) clouds, for which the time scale for the change of cloud microphysical properties is longer than the time scale for the change of environmental conditions. Additionally, since all of these methods need in-situ measurements, they have limited scope, owing to the difficulties of building statistics from short-term aircraft flights.

Here we develop a new method to estimate $s$ in clouds using ground-based remote sensing measurements. A non-drizzling
marine stratocumulus cloud is chosen to evaluate the estimation of in-cloud $s$ based on our method (equation 11), and results are compared with $s$ estimated based on in-situ measurements (equation 2). The uncertainties, limitations, and applications of our method are discussed.

## 2   In-Cloud Supersaturation Equation for Ground-Based Measurements

Our remote sensing method is adapted from McGraw (1997), in which aerosol dynamics is represented by the method of
moments and equations describing moment evolution. Here, we apply the method of moments to derive the in-cloud super-



saturation equation, starting with the definition of the $k_{th}$ radial moment ($\mu_k$) of the cloud droplet size distribution ($f(r)$),

$$\mu_k = \int r^k f(r) dr. \tag{3}$$

Differentiating the moments with respect to time gives (McGraw, 1997),

$$\frac{d\mu_k}{dt} = k \int r^{k-1} \left( \frac{dr}{dt} \right) f(r) dr. \tag{4}$$

Meanwhile, the change of droplet radius due to condensation/evaporation is governed by the diffusional growth equation (Lamb

and Verlinde, 2011),

$$\frac{dr^2}{dt} = 2Gs, \tag{5}$$

where $G$ is the growth factor depending on temperature and pressure, and $s$ is supersaturation. Combining equations 4 and 5

results in

$$\frac{d\mu_k}{dt} = kGs\mu_{k-2}. \tag{6}$$

It should be noted that the moments of $f(r)$ are related to several physical variables, such as cloud droplet number concentration

($N_d = \mu_0$) and liquid water content ($LWC = \frac{4\pi}{3}\rho_l\mu_3$ where $\rho_l$ is the density of liquid water).

Now consider the logarithmic change of $LWC$ with time for a rising parcel. The relationship between $LWC$ and $\mu_3$ leads

to

$$\frac{d\ln LWC}{dt} = \frac{1}{\mu_3}\frac{d\mu_3}{dt}. \tag{7}$$

However, the change of $LWC$ in time is also linked to the change of $LWC$ in altitude,

$$\frac{d\ln LWC}{dt} = \frac{\partial\ln LWC}{\partial z}\frac{\partial z}{\partial t} = w\frac{\partial\ln LWC}{\partial z}. \tag{8}$$

By substituting $\frac{d\mu_3}{dt}$ from equation 6 into equation 7 and combining in equation 8, we get

$$s = \frac{w}{3G}\frac{\partial\ln LWC}{\partial z}\frac{\mu_3}{\mu_1}. \tag{9}$$

Note that $\frac{\partial\ln LWC}{\partial z}$ is the gradient of $LWC$ which can be retrieved from the ground-based measurements (explained in the

following section).

If the cloud droplet size distribution is assumed to be represented by a Weibull distribution,

$$f(r) = 2\pi N_d \left( \frac{\rho_l N_d}{LWC} \right)^{2/3} r \exp\left[ -\pi \left( \frac{\rho_l N_d}{LWC} \right)^{2/3} r^2 \right], \tag{10}$$

substituting the first and third moments of equation 10 into equation 9 leads to the final equation of $s$,

$$s = \frac{w}{2\pi G}\frac{\partial\ln LWC}{\partial z}\left( \frac{LWC}{\rho_l N_d} \right)^{2/3}. \tag{11}$$





We note that assuming a different functional shape for the cloud droplet size distribution ($f(r)$) can change the expression of $s$. However, it is unclear whether there is a unified $f(r)$ for atmospheric clouds applicable for different conditions. This is a general question for cloud retrievals that is worth investigation, but it is beyond the scope of this study. We choose the Weibull distribution (equation 10) because it shows good agreement with previous in-situ measurements (e.g., Costa et al., 2000).

5 Additionally, it has a theoretical basis as it can be derived analytically either using a Brownian drift-diffusion model (McGraw and Liu, 2006), or using the maximum entropy approach constrained by surface area (Zhang and Zheng, 1994; Liu and Hallett, 1998; Wu and McFarquhar, 2018), or solving the Fokker-Planck equation (Saito et al., 2019). Recent lab experiments also show that equation 10 fits reasonably well with the cloud droplet size distribution in a turbulent cloud chamber (Chandrakar et al., in prep.). The effect and limitation of the assumed $f(r)$ on $s$ will be discussed in Section 6.

10 Our method requires $LWC$ for two adjacent layers to estimate $s$ (i.e., the $LWC$ vertical gradient). This is the major difference compared with the quasi-steady supersaturation (equation 2) where $LWC$ is only needed for one layer. In fact, $s$ estimated based on our method represents the mean $s$ between two layers and does not rely on the quasi-steady state assumption. However, if the cloud is adiabatic, meaning that $LWC$ increases linearly with height, equation 11 is dimensionally similar to equation 2: $w(\partial \ln LWC/\partial z)(LWC/N_d)^{2/3} \sim (\partial LWC/\partial z)(w/LWC^{1/3}N_d^{2/3}) \sim w/(\overline{r}N_d)$.

## 15  3  Data and Methods

We use data for a marine stratocumulus cloud observed over the Department of Energy (DOE) Atmospheric Radiation Measurement (ARM) Eastern North Atlantic (ENA) Facility on Feb. 7, 2018, during the Aerosol and Cloud Experiments (ACE-ENA) field campaign (Wang et al., 2016). The stratocumulus cloud was precipitating in the morning, but between 15 UTC and 22 UTC it had little-to-no precipitation as shown in Figure 1a. The cloud is very stable during this period, with a base height 20 of $1.23 \pm 0.03$ km and a top height of $1.52 \pm 0.02$ km. The stable cloud is also consistent with the steady meteorological conditions during that time: Surface pressure varies less than 1 hPa within seven hours (red line in Figure 1b), and surface temperature decreases only slightly from about 15 $^o$C to 13 $^o$C (blue line in Figure 1b), mainly due to the decrease of shortwave radiation (Sunset at 19 UTC). The wind is from the North and is very light near the surface with a mean value of $1.84 \pm 0.65$ m s$^{-1}$ (Figure 1c). The liquid water path and precipitable water vapor retrieved from the microwave radiometer (MWRRet; 25 Turner et al., 2007) are $54 \pm 16$ g m$^{-2}$ and $1.06 \pm 0.03$ cm, respectively (Figure 1d).

Ground-based remote sensing is used to retrieve all the variables needed to estimate in-cloud $s$ based on Equation 11, being $w$, $LWC$, and $N_d$. $w$ at each radar height is computed by taking the difference between the estimated radar reflectivity-weighted particle sedimentation speed and measured mean Doppler velocity (Kalesse and Kollias, 2013). The estimate of particle sedimentation speed at each radar time sample (every 2 sec) is obtained by mapping measured reflectivity to velocity 30 using a two-parameter Z-V power law model computed from the data within a 30-minute moving window centered on the sample. The retrieval of $LWC$ is a two-step process. The first step is to identify cloud columns in which drizzle does not dominate the reflectivity by applying two criteria. The majority of the samples in a cloud column must either have a negative Doppler spectral skewness (Luke and Kollias, 2013), assuming a positive-up velocity sign convention, or have a reflectivity of

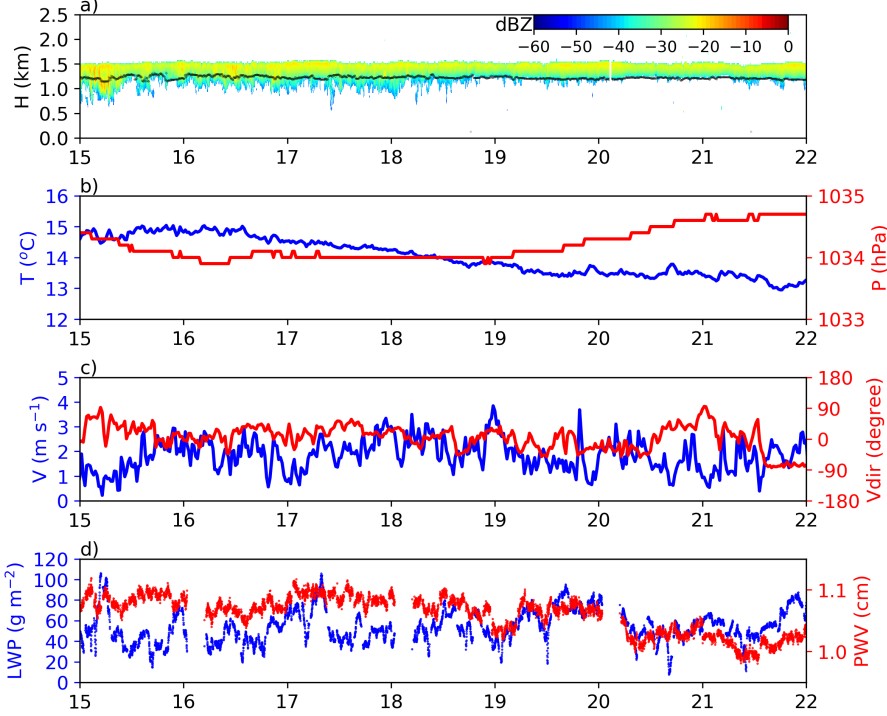

**Figure 1.** Cloud and environmental property time series observed at the Eastern North Atlantic (ENA) site on Graciosa Island (39°5′ N, 28°1′ W) on 7 February, 2018. Shown are: (a) cloud base height (black line) and Ka-band radar reflectively (colored shading), (b) surface temperature (blue line) and pressure (red line), (c) surface horizontal wind speed (blue line) and direction (red line), (d) liquid water path (blue line) and precipitable water vapor (red line).

less than -25 dBZ. The skewness and reflectivity fields used for this determination are first smoothed by a two-minute-wide box-car filter. For the columns identified, the second step is to partition the measured integrated liquid water path over the cloud vertical extent in accordance with measured radar reflectivity (Frisch et al., 1998).

Retrieving $N_d$ remains a challenge and presents larger uncertainties compared with $LWC$ and $w$ retrievals, as recently discussed in a review (Grosvenor et al., 2018). Here, $N_d$ is retrieved using ground-based micropulse lidar (MPL) measurements following Snider et al. (2017). In summary, $N_d$, cloud extinction coefficient ($\sigma$), and $LWC$ can be expressed as the zeroth, second, and third moments of cloud droplet size distributions. $N_d$ can be estimated from the lidar-derived $\sigma$ profile by assuming that the cloud droplet size distribution follows a lognormal distribution with a constant geometric standard deviation ($\sigma_g = 1.4$), and that the cloud LWC profile follows an adiabatic model. In this study, cloud $\sigma$ profiles are obtained through inversion of the lidar attenuated backscatter measurements using the Fernald (1984) method and assuming that the extinction-to-backscatter ratio for liquid droplets is 18.8 (O'Connor et al., 2004). The retrievals can be affected at cloud base by turbulent mixing and higher above cloud base by the lidar signal's quick attenuattion, so we use the mean $\sigma$ and $LWC$ over the range between 30 m and 120 m above the cloud base. Additionally, stratocumulus may have a subadiabatic $LWC$ profile, which would adversely

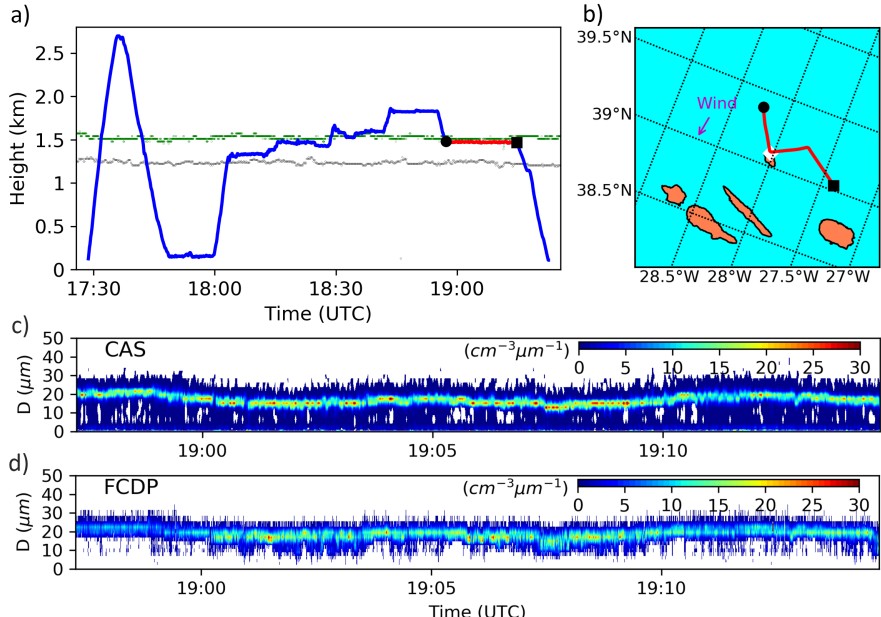

**Figure 2.** Time series of flight patterns and cloud droplet size distributions for our study period. (a) Time series of flight altitude, where black and green lines represent, respectively, the cloud base and top heights from the ground-based measurements. The red line indicates the in-cloud flight leg used in this study, and black dots represent the beginning and ending points of the selected flight leg. (b) Top view of the flight path for the selected in-cloud flight leg in (a). Ocean is bluish-green, land is coral, and the ENA Facility is marked by a white dot. Measured cloud droplet size distributions in the red flight leg are obtained from the (c) Cloud Aerosol Spectrometer (CAS) and (d) Fast Cloud Droplet Probe (FCDP).

affect the retrievals. Thus, we reduce the bias caused by subadiabatic $LWC$ profiles by scaling the $LWP$ from the adiabatic $LWC$ profile to the $LWP$ retrieved by MWRRet.

    In-situ measurements from ARM's Gulfstream-159 (G-1) aircraft on that day (Figure 2a) enable evaluating our retrieved variables and estimate of quasi-steady supersaturation (equation 2). Air vertical velocity is measured at 20 Hz by the Aircraft-

5    Integrated Meteorological Measurement System (AIMMS-20) with an uncertainty of $\pm\,0.75\,\mathrm{m\,s^{-1}}$. $N_d$ and $\overline{r}$, used to estimate $s$, are obtained from cloud droplet size distributions measured by the Cloud Aerosol Spectrometer (CAS, DMT, Inc.) and the Fast Cloud Droplet Probe (FCDP, SPEC, Inc.). The CAS measures small particles ranging from 0.5 to 50 $\mu$m in 30 size bins at 1 Hz, while the FCDP measures cloud droplets ranging from 1 to 50 $\mu$m with a resolution of about 3 $\mu$m at a frequency of 10 Hz. Because in-situ measurements from AIMMS-20, CAS, and FCDP have different sampling frequencies, we average them

10   to the lowest frequency (1 Hz) to calculate $s$, which represents the mean value over 100 m (aircraft speed 100 $\mathrm{m\,s^{-1}}$).





## 4    Results and Discussions

We use probability density functions (PDFs) of $w$, $LWC$, and $N_d$ to compare the ground-based retrievals with in-situ aircraft observations. Otherwise, it is difficult to make an apples-to-apples comparison between the two because they never sample the same cloud volume at the same time. PDFs are compared from the 105 km in-cloud flight leg (from the 17.5-min period highlighted in Figure 2a,b) with those retrieved at flight level from seven-hours of ground-based measurements, between 15 and 22 UTC. We choose that horizontal flight leg rather than all in-cloud measurements because (1) $LWC$ depends on the height above the cloud base, and (2) that flight leg was the longest, providing reliable statistics from the in-situ measurements. Note that although the flight height is very steady during the 17.5 min ($1.471\pm0.004$ m above sea level, see Figure 2a), the G-1 does not fly in one direction, as is shown in Figure 2b. The flight leg is initially 37 km from the ENA site, heading toward the site from the northwest, and then turning to the northeast. The nearest horizontal distance between the G-1 and the site is more than 1 km away, which is also why we use a statistical rather than point-to-point comparison. We assume that the stratocumulus cloud has sufficient statistical spatiotemporal homogeneity during the chosen flight period for the statistical properties to fall within a relatively narrow range. This assumption is supported by the ground-based measurements, e.g. the stable cloud base and top heights as shown in Figure 1a. A stable cloud layer also increases confidence in the determination of how far above cloud base the in-situ measurements are obtained. Additionally, cloud droplet size distributions measured by CAS and FCDP do not significantly change during the flight leg (Figure 2c,d), suggesting that the cloud is spatially homogeneous within $\sim$56 km of the site (Figure 2b).

Figure 3 shows the PDFs of four key variables used to estimate $s$. The in-situ values for $\bar{r}$, $N_d$, and $LWC$ are calculated from the cloud droplet size distributions measured by the CAS and FCDP. Because the CAS has the ability to measure sub-micron sized particles (see Figure 2c), we only use droplet diameters larger than 1.5 $\mu$m from the CAS to match the observational range of the FCDP. Results show that mean $\bar{r}$ from the CAS ($8.4 \pm 1.0$ $\mu$m) is about 1 $\mu$m smaller than that from the FCDP ($9.3 \pm 0.9$ $\mu$m) (Figure 3a), and the mean $N_d$ from the CAS (72 cm$^{-3}$) is about 15 cm$^{-3}$ larger than that from the FCDP (57 cm$^{-3}$) (Figure 3b). These differences might be due to the shattering of large cloud droplets for the CAS, resulting in reduced $\bar{r}$ and enhanced $N_d$. The retrieved $N_d$ ($71 \pm 43$ cm$^{-3}$) is significantly broader than the in-situ measurements, but it is gratifying to see that their modes are similar (Figure 3b). The mean $w$ from in-situ measurements is 0.2 m s$^{-1}$ with a standard deviation of about 0.7 m s$^{-1}$. For ground-based measurements, the retrieved $w$ at the flight level is smaller (0.04 m s$^{-1}$) with a smaller standard deviation (0.4 m s$^{-1}$) (Figure 3c). The mean $LWC$ from the CAS ($0.19 \pm 0.05$ g m$^{-3}$) and the FCDP ($0.20 \pm 0.05$ g m$^{-3}$) are similar; however, the retrieved $LWC$ is about 0.1 g m$^{-3}$ larger and with a broader PDF (Figure 3d). Note that our new method primarily depends on the vertical gradient of $LWC$ and, to a lesser degree, on its absolute value.

With the measured/retrieved variables in Figure 3, we estimate $s$ during the flight leg using the quasi-steady state assumption (equation 2) for the in-situ measurements, and $s$ at the flight level using equation 11 for the ground-based remote sensing measurements. The PDFs of $s$ are shown in Figure 4a with two noticeable features. First, although the $\bar{r}$ and $N_d$ measured by the CAS and FCDP are somewhat different (Figure 3), the PDFs of the estimated $s$ based on these two instruments are similar. This is because $s \sim (\bar{r} N_d)^{-1}$ based on equation 2 and $\bar{r} N_d$ is similar for the CAS (smaller $\bar{r}$ and larger $N_d$) and FCDP (larger





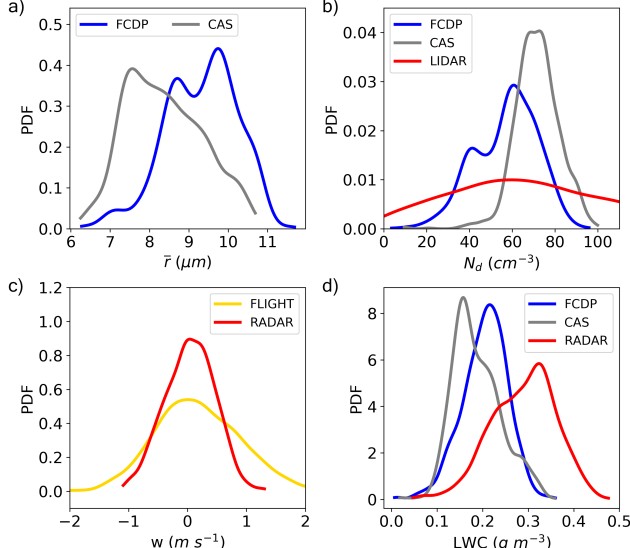

**Figure 3.** PDFs of four key variables used to estimate $s$. Shown are (a) mean droplet radius ($\bar{r}$), (b) droplet number concentration ($N_d$), (c) air vertical velocity ($w$), and (d) liquid water content ($LWC$). In-situ measured cloud microphysical properties and air vertical velocity are collected during the flight leg highlighted in Figure 2. The measurements are from the FCDP (blue), CAS (gray), and AIMMS-20 (gold). Red lines represent the variables retreived from ground-based measurements between 15 and 22 UTC. Retrieval methods are explained in the text.

$\bar{r}$ and smaller $N_d$). Second, the PDF of $s$ (i.e., supersaturation fluctuation) estimated from the retrieved variables based on the radar and lidar measurements (Figure 4a) is of the same order of magnitude as the in-situ measurements, although its full width at half maximum (FWHM) of 0.3 that is narrower than the in-situ measurements (0.6).

 With our new method, we can also estimate the profile of in-cloud supersaturation fluctuations based on the retrieved $LWC$
5 and $w$ profiles. Figure 4b shows box-and-whisker plots of estimated $s$ based on equation 11 at different heights. It is interesting to see that the $s$ fluctuations are larger either close to cloud base or at cloud top. The main reason for the larger $s$ fluctuations in these two regions likely has different root causes. Close to cloud base, $s$ fluctuations are mainly driven by stronger turbulence. This is supported by the energy dissipation rate ($\epsilon$) profile shown in Figure 4c, which is retrieved by relating the power spectrum of mean Doppler velocity computed over a 25-minute moving window to the -5/3 slope line of the Kolmogorov law of energy
10 dissipation (Borque et al., 2016). Note that turbulence having a maximum at cloud base on this day is inconsistent with the general statistical properties of $\epsilon$ in marine stratocumulus clouds, in which $\epsilon$ typically reaches a maximum close to the cloud top where turbulence is mainly driven by wind shear or radiative cooling, especially at nighttime (Wood, 2012; Borque et al., 2018). This inconsistency is believed to be because, for our particular case, the daytime cloud is thin and the wind is light; thus, turbulence in the cloud may be mainly driven by surface forcing (Borque et al., 2018). At cloud top, the large $s$ fluctuations are
15 mainly controlled by the large gradient of $LWC$ and spread therein (Figure 4d), where the large reduction of $LWC$ is likely due to the cloud-top entrainment and mixing (Mellado, 2017).



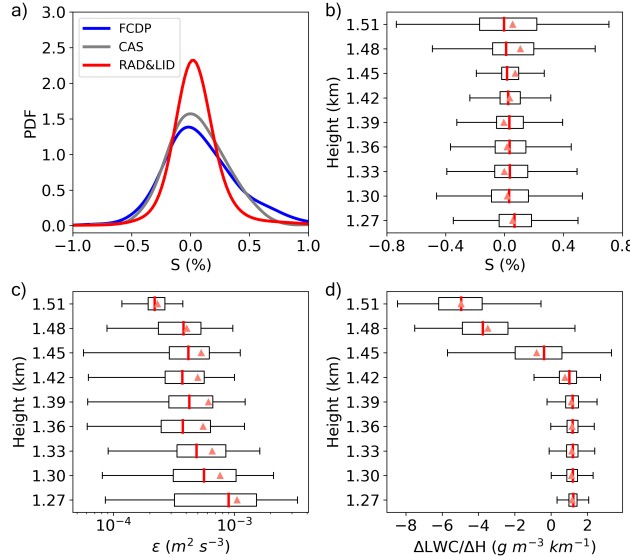

**Figure 4.** PDF of estimated $s$ and profiles of energy dissipation rate, and of the vertical gradients of liquid water content. (a) PDF of estimated $s$ from the quasi-steady state assumption (equation 2) based on the in-situ measurements from the CAS (grey) and FCDP (blue) during the flight leg highlighted in Figure 2a. PDF of estimated $s$ from remote sensing measurements (equation 11) (red) is given for the flight level between 15 UTC and 22 UTC. Box-and-whisker plots are provided for the profiles of: (b) $s$ based on equation 11, (c) the energy dissipation rate ($\epsilon$) on a log scale, and (d) liquid water content gradients ($\Delta LWC/\Delta z$) in the stratocumulus cloud between 15 UTC and 22 UTC. The box and whiskers indicate the 5th, 25th, 50th, 75th, and 95th percentiles and the means are indicated by triangles.

## 5    Uncertainties of our supersaturation estimation

Our method relies on several retrieved variables ($w$, $LWC$, and $N_d$). Although efforts have been made to improve the retrieval quality for decades, these retrieved variables still have large uncertainties that could impact the estimated supersaturation. The retrieved $LWC$ and $w$ can be affected by the existence of large drizzle or raindrops which are abundant in marine stratocu-

5    mulus clouds (Yang et al., 2018b). Minimizing the influence of drizzle on the retrieved $LWC$ and $w$ would be helpful for the estimation of supersaturation in drizzling stratocumulus clouds using our method. The retrieved $N_d$ is usually extracted from higher moments of the droplet size distribution, such as radar reflectively (6th moment) or lidar back scattering (2nd moment) such that droplet size uncertainty can cause large uncertainties in $N_d$. The uncertainty of the retrieved $N_d$ can be up to 100% which is much greater than that for the $LWC$ and $w$ retrieval (Grosvenor et al., 2018).

10    In this section, we will explore uncertainties of our estimated supersaturation fluctuations due to the uncertainties of $w$, $LWC$, and $N_d$. Specifically, we will assume that the "true" values of those variables are 0.5, 0.8, 1.2, and 2.0 times of our original retrieved values. Also, two additional retrieval methods of $N_d$ are used. While a careful investigation of which retrieval method agrees best with $N_d$ in-situ measurements is under investigation (Zhang et al., in prep.), here we only focus on the influence of different retrieval methods of $N_d$ on supersaturation estimation.



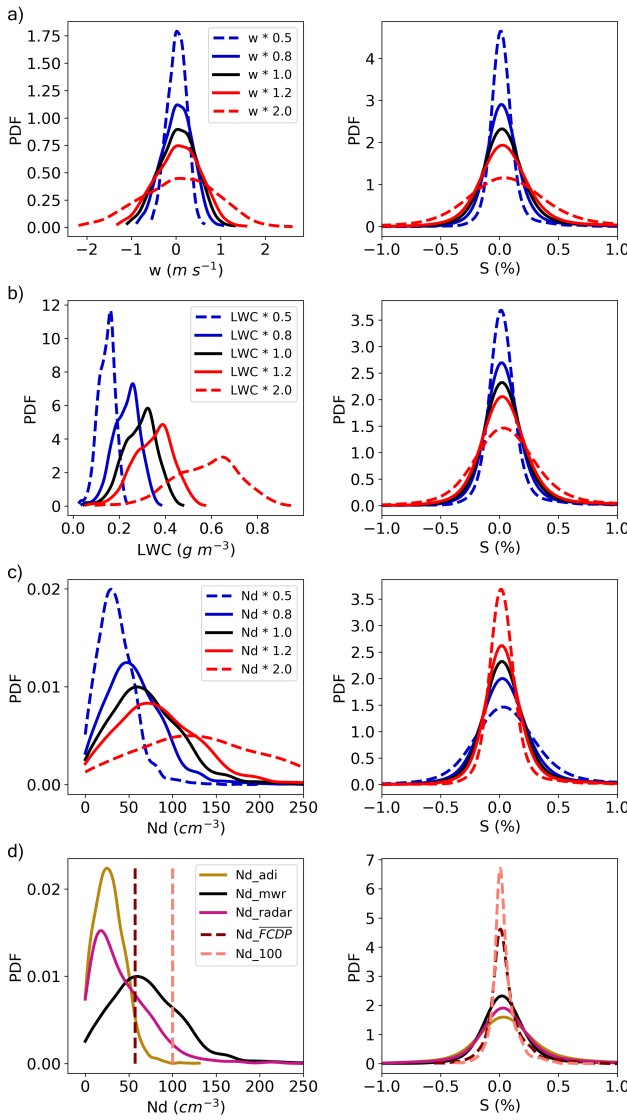

**Figure 5.** Various retrieved and modified PDFs of input properties and their corresponding the PDFs of the estimated $s$: (a) $w$, (b) $LWC$, (c) $N_d$, and (d) $N_d$ from three different retrieval methods and two constant $N_d$. Details of the modified variables and retrieval methods are described in the text.

PDFs of retrieved/modified variables ($w$, $LWC$, and $N_d$) and the corresponding estimated supersaturation are shown in Figure 5 a-c. The "truth" value of each retrieved variable is assumed to be systematically either smaller (0.5 times and 0.8 times) or larger (1.2 times and 2.0 times) than the original retrieval, while keeping other retrieved variables unchanged. It can be seen that an underestimation of $w$ and $LWC$ will lead to a larger estimated $s$ and with a broader PDF (Figure 5 a,b), while





an underestimation of $N_d$ will cause a smaller estimated supersaturation. This is because $s$ is proportional to $w$ and $LWC^{2/3}$, but inversely proportional to $N_d^{2/3}$.

Of course, all retrieved variables have uncertainties that can propagate to the estimated $s$. The total error budget of $s$ can be expressed as,

$$\left(\frac{\partial s}{s}\right)^2 = \left(\frac{\partial w}{w}\right)^2 + \left(\frac{2}{3}\frac{\partial LWC}{LWC}\right)^2 + \left(\frac{2}{3}\frac{\partial N_d}{N_d}\right)^2. \qquad (12)$$

Thus, our estimated $s$ uncertainties are more sensitive to the relative changes of $w$ compared with those for $LWC$ and $N_d$. Specifically, 20% uncertainties of $w$, $LWC$, and $N_d$ will cause a total of $(20\%^2 + \frac{4}{9}20\%^2 + \frac{4}{9}20\%^2)^{1/2} = 27\%$ uncertainty of $s$. If we assume the in-situ measurements represent "truth", the standard deviation of the retrieved $w$ is about 43% smaller, the mean retrieved $LWC$ is about 50% larger, and the uncertainty in retrieved $N_d$ can be up to 100%. So if we assume the uncertainties of the retrieved $w$, $LWC$, and $N_d$ are 43%, 50%, and 100%, respectively, the total uncertainty of the estimated $s$ is about 86%. However, we do not see such large fluctuations ($\sigma(s)/\overline{|s|}$) in Figure 4a. The reason is probably because this uncertainty assumes no correlations among the retrieved variables, and it represents a systematic bias rather than a random bias. Although such uncertainty exists in the estimated $s$, the location of the regions of larger supersaturation fluctuations should not change in Figure 4b.

The retrieved $N_d$ in the previous section uses $LWC$ profiles by scaling the integrated $LWP$ from the adiabatic $LWC$ profile to the $LWP$ retrieved by MWRRet (labeled as $N_{d\_mwr}$ in Figure 5d). Two additional retrieval methods of $N_d$ are applied here: $N_d$ is retrieved by assuming an adiabatic $LWC$ profile without the constraint of $LWP$ (labeled as $N_{d\_adi}$ in Figure 5d); $N_d$ is retrieved using $LWC$ profiles derived from the combined radar and microwave radiometer measurements following Frisch et al. (1998) (labeled as $N_{d\_radar}$). In addition, two constant values of $N_d$ are tested: mean $N_d$ from FCDP (labeled as $N_{d\_\overline{FCDP}}$) and 100 cm$^{-3}$ (labeled as $N_{d\_100}$). The PDFs of $N_d$ and the corresponding PDFs of $s$ are shown in Figure 5d. It can be seen that the FWHMs of the estimated $s$ are similar for three retrieval methods of $N_d$, with a value of 0.49 for $N_{d\_adi}$, 0.40 for $N_{d\_radar}$, and 0.37 for $N_{d\_mwr}$. But the PDFs of the estimated $s$ is much narrower for constant $N_d$, with the FWHM of 0.15 for $N_{d\_\overline{FCDP}}$ and 0.10 for $N_{d\_100}$.

## 6 Conclusions

In this study, we provide a new method to estimate in-cloud supersaturation from ground-based remote sensing measurements. Our analytical formula (equation 11) relies on the retrieved values of $w$, $LWC$, and $N_d$. The fundamental idea is that the difference in $LWC$ between two cloud layers is the result of condensation or evaporation of cloud droplets; thus, the gradient of $LWC$ together with $w$ can be used to estimate the mean supersaturation between the two layers. We evaluate our method using a stable, non-drizzling stratocumulus cloud observed over the ARM ENA Facility on Feb. 7, 2018, during the ACE-ENA field campaign. In-situ measurements from a 105 km (17.5-min) flight leg are compared to values and supersaturation estimates retrieved from ground-based measurements for the aircraft flight level. Results show that PDFs of the retrieved $w$ agree reasonably well with the in-situ measurements, while the retrieved $N_d$ has larger fluctuations, and the retrieved $LWC$





is about 0.1 $\mathrm{g\,m^{-3}}$ larger and with a broader distribution. The FWHM of the PDF of supersaturation fluctuations based on equation 11 is about half that for the quasi-steady supersaturation estimated from equation 2. We also investigate the profile of supersaturation in the stratocumulus cloud. Results show that supersaturation fluctuations are larger either at cloud base where, for this case, the eddy dissipation rate is largest due to strong turbulence, or close to the cloud top where the reduction of $LWC$

is largest due to entrainment and mixing.

Note that our analytical expression for supersaturation (equation 11) assumes that cloud droplet distribution follows a Weibull distribution (equation 10). The formula will change if a different shape is assumed for cloud droplet size distribution. We choose a Weibull distribution because it has a theoretical basis and is consistent with previous observational and laboratory studies (Costa et al., 2000; McGraw and Liu, 2006; Chandrakar et al., 2016). However, it is still unclear what the

best shape is for representing the cloud droplet distribution in atmospheric clouds over a broader range of conditions. This topic is beyond the scope of this paper but is worth investigation in the future. The uncertainty of the estimated $s$ based on our method strongly depends on the uncertainties of three retrieved variables ($w$, $LWC$, and $N_d$). Improving retrieval accuracy can increase the confidence level of the estimated $s$, as well as support other model evaluations.

The good agreement between the supersaturation estimated from ground-based measurements and from the quasi-steady

supersaturation obtained from in-situ measurements suggests that equation 11 is a suitable tool to estimate in-cloud supersaturation, which has several advantages over the quasi-steady method. First, our method does not rely on the quasi-steady state assumption, which is questionable in clean (low $N_d$) or vigorous (large $w$) clouds. In fact, supersaturation based on our method is not as sensitive to $N_d$ ($\sim N_d^{-2/3}$ in equation 11) compared with quasi-steady supersaturation ($\sim N_d^{-1}$ in equation 2), but it is sensitive to $w$ and the vertical gradient in $LWC$. Also note that retrieved $w$ is more reliable compared with retrieved $N_d$, which

is good for the estimation of $s$. Second, our method can provide profiles of supersaturation in clouds (Figure 4b), rather than point values from the in-situ measurements. Last, our method enables building statistics of supersaturation in stratocumulus clouds for various meteorological conditions from multi-year ground-based measurements.

Our method might be helpful to improve understanding of drizzle initiation in marine stratocumulus clouds. Ground-based observations at the ENA facility show that 83% of marine stratocumulus clouds are drizzling (Yang et al., 2018b). A long-

standing problem in drizzle initiation is the requirement of a mechanism to increase cloud droplet size 10-fold to become drizzle-sized. Recent laboratory and theoretical studies suggest that stochastic condensational growth due to supersaturation fluctuations could provide the link, where some lucky cloud droplets stay longer in high supersaturation regions and form drizzle drops (e.g., McGraw and Liu, 2003, 2004; Sardina et al., 2015; Chandrakar et al., 2016). Equation 11 can be used to estimate supersaturation fluctuations in atmospheric clouds for different conditions, which may be helpful to assess the

notion that drizzle formation is due to stochastic condensational growth. Finally, it should be mentioned that supersaturation fluctuation varies on spatial and temporal scales (Siebert and Shaw, 2017). If all measurements are perfect, we expect that $s$ estimated from 1-Hz in-situ measurements represents the mean over about 100 m, while $s$ estimated from ground-based measurements can represent the value in one radar sampling volume, about $10 \times 10 \times 30$ m$^3$. It is interesting to note that the radar sampling volume for one range gate is close in size to one grid box in a large eddy simulation (LES) model. This provides

an opportunity to further compare estimated and simulated supersaturation fluctuations in the future.





## Appendix A: Parameter $A$ in Equation 2

The parameter $A$ in Equation 2 follows Lamb and Verlinde (2011),

$$A = \frac{Q_1 G}{4\pi \rho_l Q_2}, \tag{A1}$$

where,

$$Q_1 = \left( \frac{l_v}{c_p T} - 1 \right) \frac{M_a g}{RT}, \tag{A2}$$

$$Q_2 = \frac{l_v^2}{M_w c_p p T} + \frac{RT}{M_w e_s}, \tag{A3}$$

$$G = \frac{\rho_l RT}{M_w D_v e_s} + \frac{\rho_l l_v}{M_w k_v T} \left( \frac{l_v}{RT} - 1 \right). \tag{A4}$$

Here $l_v$ is the latent heat of vaporization, $c_p$ is the specific heat of air at constant pressure, $M_a$ is the molar mass of air, $M_w$ is the molar mass of water, $R$ is the universal gas constant, $D_v$ is the molecular diffusion coefficient, $k_v$ is the coefficient of thermal conductivity of air, and $e_s$ is the saturated water vapor pressure at temperature $T$.

*Author contributions.* R.M. and F.Y. derived the analytical equation; F.Y. R.M. E.L. and P.K. designed research; E.L., D.Z., and F.Y. analyzed data; F.Y. and A.V. wrote the paper.

*Acknowledgements.* We thank Raymond Shaw (MTU), Yangang Liu (BNL), Katia Lamer (PSU) and Zeen Zhu (SBU) for helpful discussions. Data were obtained from the Atmospheric Radiation Measurement (ARM) Climate Research Facility, a U.S. Department of Energy Office of Science user facility sponsored by the Office of Biological and Environmental Research. All data are available from the ARM Data Discovery website (https://www.archive.arm.gov/discovery/). This work was supported by the U.S. Department of Energy (DOE) under grant DE-SC0012704.



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
