# Peer review of "A new approach to estimate supersaturation fluctuations in stratocumulus cloud using ground-based remote sensing measurements"

_Atmospheric Measurement Techniques, 2019_

## Referee Comment (RC2) · Anonymous Referee #3 · 22 Jul 2019

The comment was uploaded in the form of a supplement:
https://www.atmos-meas-tech-discuss.net/amt-2019-222/amt-2019-222-RC2-supplement.pdf
* * *

---

## Author Comment (AC1) · 12 Sep 2019

**Response to the reviewer's comments:**

The manuscript describes a new approach to retrieve in-cloud water vapor supersaturations based on radar and lidar measurements. The approach is applied to data of the ACE-ENA field campaign, including a comparison with in-situ measurements and an assessment of uncertainties. The manuscript is overall interesting, well-written, and I support its publication in Atmospheric Measurement Techniques once my minor concerns are addressed.

We would like to thank the reviewer for careful reading of the manuscript and helpful comments.

P. 1, l. 13: The supersaturation in clouds does not only depend on the updraft velocity of a cloud (to which "calm" and "energetic" most likely refer to) but also on the cloud microphysical composition (see, e.g., Grabowski and Wang (2013, doi: 10.1146/annurev-fluid-011212-140750)).

We modify the text based on the reviewer's comment:

" **Supersaturation, which plays a crucial role in cloud droplet formation and growth, depends on air vertical velocity and can also be adjusted by cloud microphysical composition (Lamb and Verlinde, 2011; Grabowski and Wang, 2013).**"

P. 2, ll. 1 – 2: The adverb "simultaneously" describes a process happening at the same time. It is, however, also important that the measurements are co-located in physical space.

Agreed. We modify the text as,

"If we can simultaneously measure water vapor pressure ($e$) and temperature ($T$) **at the same location**, …"

P. 2, ll. 11 – 12: It is misleading to write about a specific location of the effective s determined by the CDNC closure method ("s in the CCN counter"). To my understanding, the s describes the maximum s at cloud base where it is able to activate the number of CCN that are measured in the CCN counter.

Agreed. To make the sentence clear, we modify it as,

"For the CDNC closure method, s in the CCN counter is  **used to describe the maximum s at cloud base** if  **the number of activated CCN in the counter is similar to that of cloud droplets**."

P. 2, ll. 19 – 21: Since Eq. 2 is not an integral, changes in the microphysical composition in time and space are not important. However, I agree that once Eq. 2 is applied to a larger volume (i.e., it is integrated), changes in the microphysical composition will matter, especially if they are fast as it is the case for small Nd and large w as stated correctly by the authors. Therefore, the sentence should read as: "[…] for which the time scale for the change of cloud microphysical properties is shorter than the time scale for the change of environmental conditions."

We still think that the quasi-steady state assumption does not hold when the time scale for the change of cloud microphysical properties (which is the phase relaxation time) is **longer (not shorter)** than the time scale for the change of environmental conditions (e.g., the mixing time). The phase relaxation time describes how quickly a cloud can response to the change of environmental conditions. A longer phase relaxation time means the cloud responds slower to the environmental change, in other words, cloud needs a longer time to return to the quasi-steady state. In order to eliminate the potential confusion, we modify the text as,

"However, this assumption might not be valid in clean (low $N_d$) and/or vigorous (large w) clouds,  **for which clouds need a longer time to return to the quasi-steady state for the change of environmental conditions**."

P. 3, Eq. 4: It might be helpful to give the reader a hint on why one can neglect the time dependency of f(r) during integration (although it becomes apparent after some thinking).

The reviewer raises a subtle point! Equation 4 follows from the continuity equation for particle number, which is conserved for condensation/evaporation, for which an integration by parts uses the fact that f(r) vanishes at the limits of integration. We add more description to the manuscript:

"**It should be mentioned that equation 4 follows from the continuity equation for particle number, which is conserved for condensation/evaporation, for which an integration by parts uses the fact that f(r) vanishes at the limits of integration (McGraw and Wright, 2003)**."

P. 3, Eq. 8: By restricting the temporal change of LWC to changes in height and then vertical velocity (Eq. 8), other important processes affecting the supersaturation (foremost entrainment and mixing processes) are neglected. I believe that this simplification is valid in stratocumulus, in which entrainment and mixing are less important than in cumulus clouds. And in fact, the authors have chosen a relatively low turbulent stratocumulus cloud (p. 4, ll. 19 – 20) in which the inherent assumptions of Eq. 8 are probably fulfilled. However, I strongly recommend the authors to comment more on the implications of Eq. 8, especially the neglected cloud processes, to account for potential other applications of this approach.

Very good point. We add more text on the limitations of Eq. 8.

" **If the lateral entrainment and mixing are not the main processes affecting supersaturation, which is likely to be true for stratocumulus clouds,** the change in LWC with time  **can** also **be** linked to the change of LWC in altitude,"

P. 5, l. 8: It probably will not change the results significantly, but why do the authors not assume a Weibull distribution here (as done in Sec. 2)?

The method we use to retrieve cloud droplet number concentration is based on Snider et al. (2017), in which a lognormal size distribution is assumed. If we use a Weibull distribution, the retrieved cloud droplet number concentration will be 25% larger. Such difference is smaller than using different retrieval methods, as shown in Figure 5d. Using the exact same method and size distribution used in Snider et al. (2017) is helpful to compare the retrieved cloud droplet number concentration with other studies, and it will be easier to extend the application of equation 11 for people who only use retrieval products. To make the text clear and consistent in the manuscript, we add more discussion,

"**It should be mentioned that the method we use to retrieve cloud droplet number concentration is based on Snider et al. (2017), in which a lognormal size distribution is assumed. If we use a Weibull distribution, the retrieved cloud droplet number concentration will be 25% larger (detailed in the Appendix B). Such difference is smaller than using different retrieval methods, as shown in Figure 5d. Using the exact same method and size distribution used in Snider et al. (2017) is helpful to compare the retrieved cloud droplet number concentration with other studies, and it will be easier to extend the application of equation 11 for people who only use retrieval products.**"

"**Appendix B: Retrieving cloud droplet number concentration**

**Based on Snider et al. (2017), the cloud droplet number concentration can be retrieved from the lidar backscatter coefficient ($\sigma$) and liquid water content ($q_l$),**

$$N_d = \frac{2e^{3\sigma_x^2}\rho^2}{9\pi}\frac{\sigma^3}{q_l^3},$$

**where the cloud droplet size distribution is assumed to be lognormal and have a standard deviation of $\sigma_x = \ln 1.4$. If we assume the cloud droplet sizes to follow a Weibull distribution (Equation 10), the cloud droplet number concentration has a similar relationship between $\sigma$ and $q_l$ but a different prefactor,**

$$N_d = \frac{\rho^2}{8}\frac{\sigma^3}{q_l^3},$$

**Specifically, the retrieved cloud droplet number concentration using a Weibull distribution is 25% larger than if a lognormal distribution was used.**"

 s fluctuations at the cloud base could also arise from changes in cloud base height, and therefore differences in thermodynamics and not turbulence.

The reviewer makes a very good point. But we think the turbulent effect is stronger than the thermodynamic effect under most of the conditions in our case, because: (1) the cloud-base height is roughly constant around that time as shown in Figure 1; (2) when we calculate supersaturation, we only consider the case when both layers (upper layer and lower layer) have the retrieved LWC values—if the cloud-base height is higher such that only one layer (upper layer) has a retrieved LWC, we do not use the profile; (3) Figure 4 shows that the gradient of LWC is small at cloud base, compared with that at cloud top, suggesting that the broader distribution of s fluctuation at cloud base is unlikely mainly due to the narrower distribution in LWC gradient (thermodynamic effect). But it is still possible that the thermodynamic effect is stronger than the turbulent effect at the cloud base in some local regions. We add some discussion to the manuscript,

"…**It should be mentioned that a larger fluctuation in s at cloud base could also arise from changes in cloud-base height. However, this thermodynamic effect is unlikely the main cause in our case because the gradient of LWC close to cloud base is relatively small, as shown in Figure 4d.**"

 "[…] large drizzle and raindrops which are abundant in marine stratocumulus clouds […]": This means that all marine stratocumulus clouds are drizzling or raining which is not true. In fact, the authors state that the analyzed stratocumulus cloud does not precipitate (p. 4, ll. 18 – 20).

Agreed. To make the manuscript accurate and consistent, we modify it as,

"…large drizzle or raindrops which are  **frequently observed** in marine stratocumulus clouds…"

 There are some typos in the equations. Check Korolev and Mazin (2003, doi:10.1175/1520-0469(2003)060<2957:SOWVIC>2.0.CO;2) for details. Please also check if these errors affected the results of the manuscript. I state the corrected equations below (changes are highlighted in red):

Thank the reviewer for checking the equation carefully. The equation is correct because we follow the equations used in Lamb and Verlinde (2011). The difference is that Lamb and Verlinde use a mole-based unit, while Korolev and Mazin use a mass-based unit. For example, the unit of $l_v$ is J/mol in Lamb and Verlinde (2011), but it is J/kg in Korolev and Mazin (2003). To make it clear, we added a statement to the manuscript:

"**Note that the unit here is mole-based, i.e., the units for $l_v$ is J/mol (e.g., not J/kg).**"

P. 3, l. 1: It should read "k-th" or "k$^{th}$", but not "k$_{th}$".

Corrected in the manuscript.

P. 3, Eq. 8: The equation should read: $\frac{dlnLWC}{dt} = \frac{\partial lnLWC}{\partial z}\frac{dz}{dt} = w\frac{\partial lnLWC}{\partial z}$, with a total derivative of z.

Corrected in the manuscript.

P. 7, l. 8: The flight was probably at $1.471 \pm 0.004$ km above sea level and not $1.471 \pm 0.004$ m.

Corrected in the manuscript.

P. 11, l. 7: Add parentheses to all squared percentage values, e.g., $(20\ \%)^2$.

Corrected in the manuscript.

---

## Author Comment (AC2) · 12 Sep 2019

**Response to the reviewer's comments:**

Supersaturation (s) in cloud is hard to measure even with an in-situ instrument that aims at measuring it directly in cloud, mainly due to the difficulty of measuring humidity in cloudy environment. Estimating it utilizing ground remote sensing instruments would be even more challenging. The authors in this manuscript propose a new approach to estimate s in such a way. The formulation of the equation that derives s seems to be done solidly and the key variables in the equations seem to be obtainable from radar and lidar output with some assumptions. If s can be estimated reasonably well by the proposed method, it would benefit a lot on the effort to estimate cloud supersaturation profile in a continuous manner, which would be impossible from aircraft measurements that are inherently episodic and expensive. Despite such enormous benefits, however, the uncertainty of estimated s with ground remote sensing data seems insurmountably high, according to the results described in this manuscript. Not only the PDF of the estimated s is much narrower than those obtained from in-situ cloud microphysics measurement but also the mode values of s do not seem to match well among each other, about which the authors did not mention anything. Moreover, there is no guarantee that the estimated s from the in-situ cloud microphysics measurement represent true s in cloud because such estimation itself is based on the big "quasi-steady state assumption," rather than from direct and correct humidity measurement. So even if they do match well, that does not mean that the estimated s from ground remote sensing data represent the true s. Therefore, I am not sure if realistic in-cloud supersaturation values are obtainable from this approach. Perhaps this approach of estimating s can still be very useful as a way to get a relative measure of in-cloud supersaturation. In that sense, I urge the authors to do s estimations in some other clouds using the same approach and see how much they are different from the one presented in this manuscript. Matching in situ cloud microphysics measurement may not be available for these other clouds but that is ok. Here the purpose is to demonstrate the capability of this approach to estimate different s distribution for different clouds.

We would like to thank the reviewer's constructive comments. We agree with the reviewer that, even when supersaturation fluctuations estimated from our method agree with that estimated from in-situ measurements, we still don't know the truth, and we believe that nobody knows the truth at present. As the reviewer states "this approach of estimating s can still be very useful as a way to get a relative measure of in-cloud supersaturation"—this is also the idea we want to express in the manuscript. For example, Figure 4b shows the s profiles in stratocumulus clouds. The most important and useful information is **not** to say whether s fluctuation is 0.3% or 0.4% (which depends on those retrieval variables), **but** that there are relatively narrower or broader regions of s fluctuation. It is interesting and useful to investigate the s distribution for different clouds in the future, but it is beyond the scope of the scope of this manuscript for the following reasons. There are two main points to be made.

First, we think it is important to convey to people that there is a new method and that it works by evaluating the retrieved variables and estimated supersaturations with in-situ measurements before we extend this method to estimate supersaturation under other cloud conditions. To do so, we went through all cases during the ACE-ENA field campaign to find that the case we present

in this study is the best one and the only one suitable for evaluation due to the stable cloud base and long flight leg. "Matching in situ cloud microphysics measurement may not be available for these other clouds but that is ok" is meaningful after we show some evidence that our method compares reasonably well with in-situ measurements, which is the scope of this study.

Second, we agree with the reviewer that it will be very interesting to "look at other clouds using the same approach and see how much they are different from the one presented in this manuscript." However, one should be very careful to estimate supersaturation for different cloud conditions. The reason is that supersaturation estimated based on our method strongly depends on the retrieved variables, and the retrieved variables might have different uncertainties under different cloud conditions. The advantage of the case we choose in this study is that the environmental conditions are very stable (Figure 1), which is the best case for retrieval and evaluation compared with other days. Further, as mentioned by the other reviewer, use of a stratocumulus cloud minimizes the effects of lateral entrainment and mixing, which may not be true for other cloud types (except for their more-spatially limited cores).

In summary, we agree with the reviewer that supersaturation estimated based on our method might not "represent the true s", because nobody knows what truth is. But it is a method to get "a relative measure of in-cloud supersaturation". We believe that there could be rich scientific implications of this study, as the reviewer mentioned, to look at the "supersaturation in some other clouds". It is interesting, but more time and effort is needed. In this study, we choose one aspect of the scientific application: the supersaturation fluctuation profile in the stratocumulus cloud (Figure 4), which is not achievable from in-situ measurements. The relative broader regions of s fluctuation suggest the important roles of dynamics and thermodynamics.

We add some discussion in the manuscript to highlight the merit and limitation of this study,

"…**It should be mentioned that our approach of estimating s cannot obtain the true supersaturation at the current stage due to (1) the difficulty of directly measuring s for evaluation and (2) the uncertainty in retrieved variables used in our method, but it still can be very useful to get a relative measure of in-cloud supersaturation. For example, for the profiles in Figure 4b, the most important and useful information is to say where there are relatively narrower or broader regions of s fluctuation, not whether the s fluctuation is 0.3% vs. 0.4%, as the latter would require exact accuracy in the retrieved variables. It is interesting and useful to investigate the s distribution for different clouds in the future.**…"

" **s estimated based on this method indicates the maximum supersaturation at cloud base for droplet activation and does not represent the real-time in-cloud supersaturation fluctuation**…"

P4, L26: No explanation is given on ground remote sensing instruments. Add brief explanation.

We add more discussion about the remote sensing instruments in the manuscript.

"**The site's zenith-pointing Ka‑band cloud radar (KAZR), ceilometer, micropulse lidar (MPL), and microwave radiometer were used to derive the information on vertical air velocity, liquid water content, and cloud droplet number concentration. The vertical resolution of the KAZR is 30 m (one range gate), and for the ceilometer and MPL is 15 m. The temporal resolutions of the KAZR, ceilometer, and MPL are 2 s, 16 s, and 10 s respectively. At the distance of the observed targets, the radar and lidar beams are, respectively, about 3 m and 2 mm wide. The microwave radiometer employs three receiver channels operating at 23.84, 31.4, and 90 GHz, providing liquid water path estimates at a temporal resolution of 3 s. Specifically**,…"

P5, L8: Nd is estimated using a lognormal distribution assumption. This distribution is similar but different from the Weibull distribution that was originally used for formulating s equation (Eq. 11). Explain the effect of such change.

The method we use to retrieve cloud droplet number concentration is based on Snider et al. (2017), in which a lognormal size distribution is assumed. If we use a Weibull distribution, the retrieved cloud droplet number concentration will be 25% larger. Such difference is smaller than using different retrieval methods, as shown in Figure 5d. Using the exact same method and size distribution used in Snider et al. (2017) is helpful to compare the retrieved cloud droplet number concentration with other studies, and it will be easier to extend the application of equation 11 for people who only use retrieval products. To make the text clear and consistent in the manuscript, we add more discussion,

"**It should be mentioned that the method we use to retrieve cloud droplet number concentration is based on Snider et al. (2017), in which a lognormal size distribution is assumed. If we use a Weibull distribution, the retrieved cloud droplet number concentration will be 25% larger (detailed in the Appendix B). Such difference is smaller than using different retrieval methods, as shown in Figure 5d. Using the exact same method and size distribution used in Snider et al. (2017) is helpful to compare the retrieved cloud droplet number concentration with other studies, and it will be easier to extend the application of equation 11 for people who only use retrieval products.**"

"**Appendix B: Retrieving cloud droplet number concentration**

**Based on Snider et al. (2017), the cloud droplet number concentration can be retrieved from the lidar backscatter coefficient ($\sigma$) and liquid water content ($q_l$),**

$$N_d = \frac{2e^{3\sigma_x^2}\rho^2}{9\pi}\frac{\sigma^3}{q_l^3},$$

**where the cloud droplet size distribution is assumed to be lognormal and have a standard deviation of $\sigma_x = \ln 1.4$. If we assume the cloud droplet sizes to follow a Weibull distribution (Equation 10), the cloud droplet number concentration has a similar relationship between σ and q$_l$ but a different prefactor,**

$$N_d = \frac{\rho^2}{8} \frac{\sigma^3}{q_l^3},$$

**Specifically, the retrieved cloud droplet number concentration using a Weibull distribution is 25% larger than if a lognormal distribution was used.**"

P9, L11-12: 20% overestimation is not exactly meaning that the true value is 0.8 times the retrieved value. Just use one metric to avoid confusion.

To make the text clearer, we modify it as,

"Specifically, we will assume that the "true" values of those variables, respectively, are **0.5x, 0.8x, 1.2x, and 2.0x** our original retrieved values."

P10, L4-5: Similarly confusing. We do not usually say "overestimate 0.5 times." Just say 0.5 times the true value (underestimation) or 1.2 times the true value (overestimation).

To make the text clearer, we modify it as,

"The "true" value of each retrieved variable is assumed to be systematically smaller (**0.5x or 0.8x**) or larger (**1.2x or 2.0x**) than the original retrieval."